# Investigation on the High-Temperature Oxidation Resistance of Ni-(3~10) Ta and Ni-(3~10) Y Alloys

**Hang Duan [1,2,\*]** , **Yan Liu [1,\*]** , **Tiesong Lin [3]** , **Hui Zhang [1]** and **Zhengren Huang [1,\*]**

[1]  State Key Laboratory of High Performance Ceramics and Superfine Microstructures, Shanghai Institute of Ceramics, Chinese Academy of Sciences, Shanghai 201800, China; zhanghui@mail.sic.ac.cn

[2]  University of Chinese Academy of Sciences, Beijing 100049, China

[3]  State Key Laboratory of Advanced Welding and Joining, Harbin Institute of Technology, Harbin 150000, China; tiesonglin@hit.edu.cn

\*  Correspondence: duanhang@shanghaitech.edu.cn (H.D.); stony2000@mail.sic.ac.cn (Y.L.); zhrhuang@mail.sic.ac.cn (Z.H.); Tel.: +86-021-69906056 (H.D. & Y.L. & Z.H.)

**Abstract:** Ni-(3~10) Ta and Ni-(3~10) Y alloys were fabricated by vacuum arc melting. The oxidation resistance of the alloys was studied by cyclic and isothermal oxidation tests at 800 °C in static air. The present work focused on the investigation of the effects of the alloying elements (Ta and Y) on the oxidation behavior of Ni-based alloys. The oxidation behavior of alloys was evaluated by mass gain, composition, as well as the microstructure of oxidized products. The experimental results indicated that Ta at a low content (3 wt %) had a positive role in enhancing oxidation resistance by decreasing the oxygen vacancy concentration of the oxide layer to prevent the inward diffusion of oxygen during oxidation, and the mass gain decreased from 2.9 mg·cm$^{-2}$ to 1.7 mg·cm$^{-2}$ (800 °C/200 h), while Y (3~10 wt %) degraded the oxidation resistance. However, it is worth mentioning that the pinning effect of $Y_2O_3$ increased the adhesion between the substrate and oxide layer by changing the growing patterns of the oxide layer from a plane growth to fibrous growth. Among the results, the bonding of the substrate and oxide layer was best in the Ni-3Y alloys.

**Keywords:** Ni-based alloys; oxidation resistance; cyclic oxidation test; Ta and Y

## 1. Introduction

Ni-based alloys have remarkable mechanical properties and excellent oxidation resistance at high temperatures. These alloys are used extensively for high-temperature industrial applications, such as aerospace systems, gas turbine blades, and interconnects of solid oxide fuel cell (SOFC) [1–3]. In these applications, with the working temperature of SOFC decreasing to an intermediate temperature (600–800 °C), metallic materials (Fe-based, Ni-based, and Cr-based alloys) as interconnects become feasible [4–6]. However, Cr in Ni-based alloys can result in Cr volatilization and deposition on the cathode, which can significantly degrade SOFC performance [7,8]. Because of the oxidation product, $Al_2O_3$, Al is also known to enhance the oxidation resistance [3]. However, the high-temperature conductivity of $Al_2O_3$ is extremely poor, which would reduce electrical conductivity. Interconnects must work at high temperatures without a decline in electrical properties and oxidation resistance. To overcome this issue, our previous work completed the evaluation of TiN/Ni cermet as interconnects for solid oxide fuel cells [9]. To further improve the high-temperature oxidation resistance, we considered strengthening the oxidation resistance of metal matrix Ni in the cermet by adding Ta or Y alloying elements.

The composition of Ni-based alloys is a significant factor in controlling oxidation resistance [10,11]. Previous researchers have studied oxidation behavior and tried to find ways to improve the oxidation

resistance of Ni-based alloys, and most concentrated on the effects of multiple metal elements on the cyclic oxidation of Ni-based alloys [12]. It is not easy or accurate to analyze the specific effect of each element, due to the diversity of added elements. However, in the alloys investigated in this paper, there are only two components. In a past research report, Jalowicka et al. showed that a small addition of Ta (0.3 at %) in Cr–Co–Ti complex superalloys at 1050 °C improved high-temperature oxidation resistance, whereas the oxidation resistance was reduced when the concentration was higher (3 at %) [13]. Klein et al. reported that the oxidation resistance of Co-based alloys decreased under an addition of 2 at % (~5 wt %) of Ta [14]. Elliott et al. suggested that Ni-10Cr alloys including Nb and, particularly Ta, behaved better during both cyclic and isothermal oxidation [15]. Moreover, Chicardi et al. reported that Ta could positively enhance the oxidation resistance of (Ti, Ta) (C, N)-based composites by forming $Ta^{5+}$ in the oxide layer to reduce the oxygen vacancy concentration [16,17]. Chuprina et al. analyzed the oxidation kinetics and the composition of the oxide layer of an intermetallic compound, NiTa, at temperatures ranging from 600 °C to 1000 °C, but no single element, such as Ni or Ta, was given for comparative experiments [18]. Wu et al. showed that a small addition of Y (0.1 and 0.3 at %) in Al-rich TiAl alloys was effective in improving the adhesion between the substrate and the oxide layer [19,20]. Moosa et al. reported that, under 0.1 wt % and 0.3 wt %, a small addition of Y could increase the oxidation rate of Ni at 500 to 600 °C, but reduce it at 700 °C to 900 °C. However, this experiment lacked sufficient data, such as X-ray diffraction (XRD) patterns and cross-section micrographs of the oxide layer, to better prove the conclusion [21]. Zhao et al. reported that in TiAl alloys containing high Nb, only a suitable amount of Y addition (0.3 at %) could enhance long-term oxidation resistance under both cyclic and isothermal oxidation. Either excessive or deficient addition of Y was detrimental to oxidation resistance [22]. Therefore, we believe that Ta and Y are effective elements to improve the oxidation resistance of pure Ni, and are worthy of study.

In this work, Ni-based alloys with a small addition of Ta or Y element were prepared by vacuum arc melting. Cyclic oxidation tests were performed in static air at 800 °C. After testing, the oxide scales of the Ni-based alloys were examined by X-ray diffraction and scanning electron microscopy (SEM). Furthermore, the effect of Y or Ta on the oxidation resistance of Ni-based alloys was also compared with pure Ni samples. Through experimental results, the Ni–Me (Me = Ta or Y) alloys with the best oxidation resistance could be chosen as a metal matrix of TiN/Ni–Me cermet applied for intermediate-temperature solid oxide fuel cell interconnects, which is a future research direction. Therefore, this work is of great importance to our future research.

## 2. Materials and Methods

### 2.1. Samples Preparation

Granular metal particles (Ni, Ta, and Y) with an average size of $\phi$3 mm $\times$ 4 mm (99.9% in purity, Yipin Chuancheng Technology Co., Ltd., Beijing, China) were used for melting stock in the experiments. The chemical compositions of the Ni and Ni-based alloys selected for this study are shown in Table 1. The samples were prepared by using vacuum arc melting (DHL-300, Keyi Co., Ltd., Shenyang, China), with argon gas as the protective gas and a working electric current of about 220 A, melting them several times to improve homogeneity. After melting, the ingots were cut into several bars with a size of 20 mm $\times$ 4 mm $\times$ 4 mm using a CNC wire cutting machine. The bar surfaces were polished with SiC sandpaper to 1200 # to remove the surface oxide and keep the surfaces smooth. Afterwards, $Al_2O_3$ abrasive paste with different particle sizes was used to buff them, and then P-1-type polishing machine (Shanghai Electric Machinery Co., Ltd., Shenyang, China) was used to polish them and make flat mirror surfaces. Finally, the samples were cleaned with acetone for 30 min using an ultrasound method and then dried for the characterization and oxidation tests.

**Table 1.** Different compositions of Ni-based alloys selected for this study (wt %).

| Alloys | Ni | Ta | Y |
|--------|-----|----|----|
| Pure Ni | Bal. | - | - |
| Ni-3Ta | Bal. | 3 | - |
| Ni-6Ta | Bal. | 6 | - |
| Ni-10Ta | Bal. | 10 | - |
| Ni-3Y | Bal. | - | 3 |
| Ni-6Y | Bal. | - | 6 |
| Ni-10Y | Bal. | - | 10 |

## 2.2. Oxidation and Characterization

Phase analysis was conducted by XRD using a Guinier-Hagg camera (Jungner Instrument, Expectron XDC-1000, Solna, Sweden) with Cu kα radiation. Microstructures were analyzed using a scanning electron microscope (SEM, S-4800, HITACHI, Tokyo, Japan) equipped with an energy dispersive spectrometer (EDS). The cyclic oxidation experiments were performed by repeatedly exposing the samples to static air at 800 °C for a total of 200 h. An electronic microbalance (Shanghai Shunyu Hengping Scientific Instrument Co., Ltd., Shanghai, China), with an accuracy of $10^{-4}$ g, was used to weigh the mass gain of samples every 20 h until 200 h.

## 3. Results and Discussions

### 3.1. Properties of the Ni-Based Alloys

Figure 1 shows the XRD patterns of Ni-based alloys obtained by vacuum arc melting. As the image shows, all the samples with different compositions include two individual phases without pure Ta or Y phases, Ni and $Ni_xMe_y$ (Me = Ta or Y), indicating that the doping of Ta and Y was uniform and fully melted.

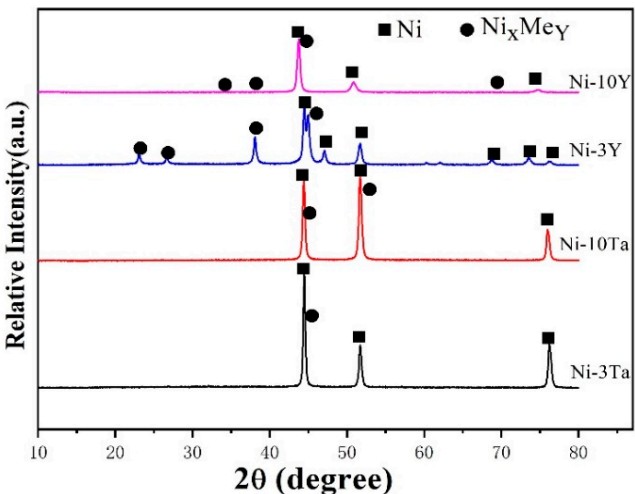

**Figure 1.** XRD patterns of the Ni-based alloys with different compositions.

Figure 2 exhibits the microstructures of the Ni-based alloys with different doping elements. The different addition content had no great influence on the micrographs of Ni–Me alloys. Thus, Figure 2 only shows the microstructures of the two samples. As can be concluded from the SEM images of the Ni-3Ta alloys (Figure 2a), the Ni matrix was distributed with irregular dendrites phases, which EDS analysis identified as $Ni_2Ta$. As for the Ni-3Y alloys (Figure 2b), discontinuous island phase A and irregular particle phase B were dispersed in the Ni matrix. According to EDS analyses, area A and B were made up of $Ni_4Y$ and $Ni_3Y$, respectively.

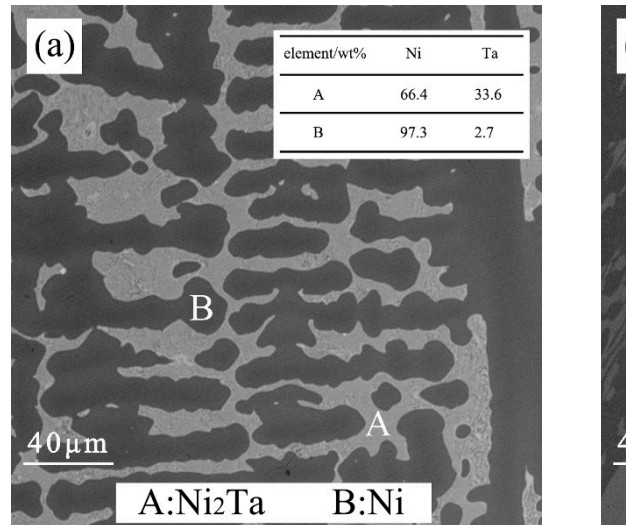
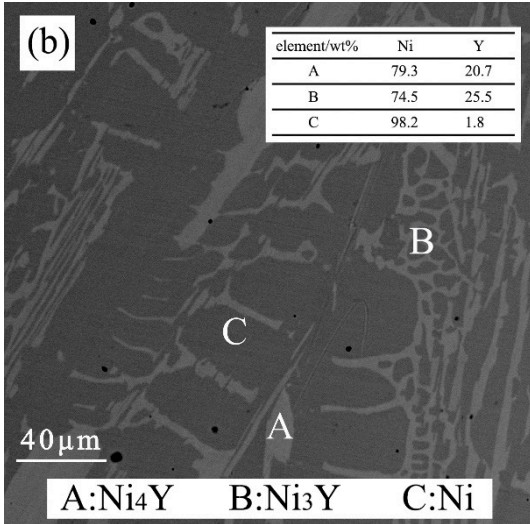

**Figure 2.** SEM images of the Ni-based alloys: (**a**) Ni-3Ta, (**b**) Ni-3Y.

The Ta and Y were 2.85 wt % and 2.92 wt %, respectively, in the Ni-3Ta and Ni-3Y alloys by using X-ray fluorescence analysis, which showed that the loss was low during smelting, and the elements were fully doped. The loss was also low for a high Me addition. The melting point of the Ni-based alloys was determined by differential scanning calorimetry ranges, from 1200 °C to 1410 °C.

### 3.2. High-Temperature Oxidation Resistance of the Ni-Based Alloys

Figure 3 presents the evolution of weight gain as a function of time for the Ni-based alloys with different compositions, and all results satisfy the parabolic kinetics law [23] described by:

$$(\Delta W/A)^2 = Kt \tag{1}$$

where $\Delta W$ is the mass gain, $t$ is oxidation time, $A$ is sample surface area, and $K$ is the oxidation rate constant, which reflects the speed of the oxidation process. Figure 4 shows the squares of mass gains as functions of time at 800 °C for the Ni-based alloys. The parabolic rate constant ($K$) calculated from the slope of the linear fitting in Figure 4, coefficient of determination ($r$), and mass gain of Ni-based alloys oxidized at 800 °C for 200 h are listed in Table 2. The coefficients of determination ($r$) are all better than 0.99, which shows that the oxidation of the alloys followed a parabolic mechanism. As for the Ni-(3~10) Ta alloys (Figure 4a), it can be clearly seen that the $K$ for pure Ni is higher than for Ni-3Ta alloys, and lower than for Ni-6Ta and Ni-10Ta alloys, which indicated that, only under appropriate additional concentration, Ta had a positive role in enhancing the oxidation resistance of alloys. As Ta content decreases from 10 wt % to 3 wt %, the weight gain reduces from 3.8 mg·cm$^{-2}$ to 1.7 mg·cm$^{-2}$. However, as with Ni-(3~10) Y alloys, the $K$ for Ni-Y alloys with different compositions are always higher than for pure Ni. Moreover, after oxidation for 200 h, the mass gain of Ni-(3~10) Y alloys is approximately three and six times higher than of pure Ni samples. As shown in Table 2, the $K$ value for Ni-3Ta alloys was approximately one and two orders of magnitude lower than for the Ni-(3~10) Y alloys. The addition of Y in the Ni-based alloys did not inhibit but promoted the oxidation process.

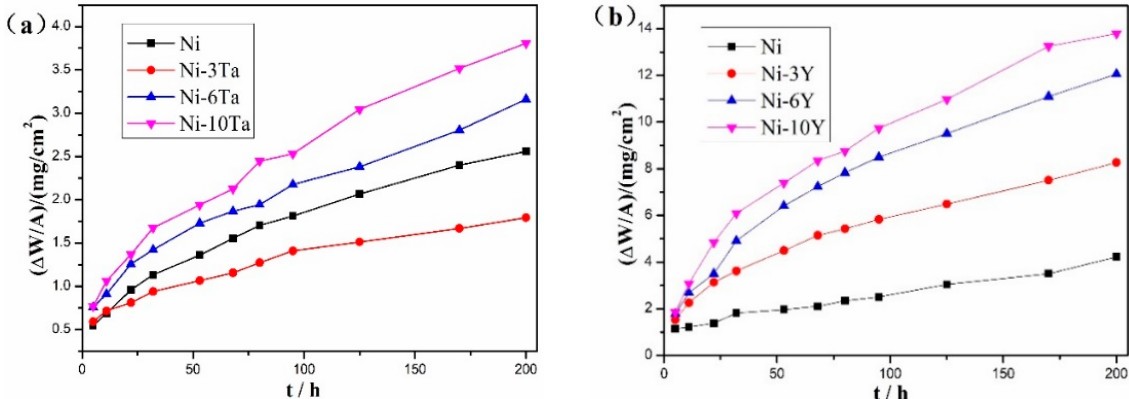

**Figure 3.** The mass gain of Ni-based alloys as a function of oxidation time at 800 °C: (**a**) Ni-(3~10) Ta alloys and (**b**) Ni-(3~10) Y alloys.

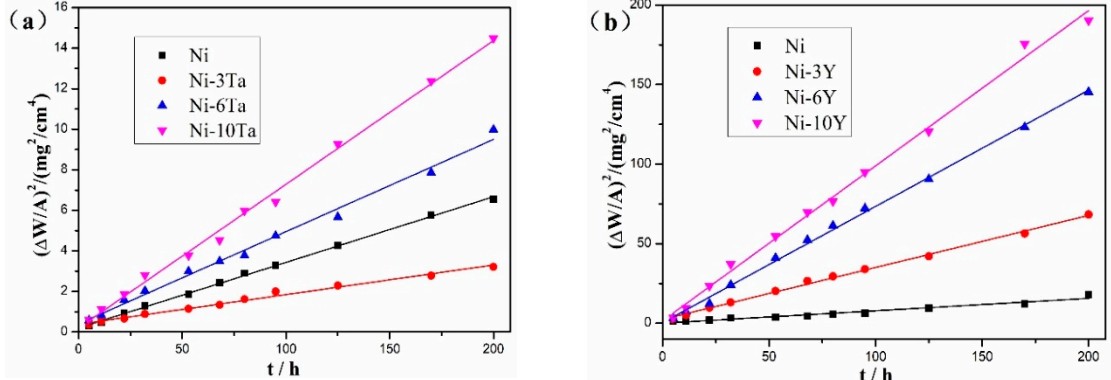

**Figure 4.** The squares of mass gain of Ni-based alloys as a function of oxidation time at 800 °C: (**a**) Ni-(3~10) Ta alloys and (**b**) Ni-(3~10) Y alloys.

**Table 2.** Values of the parabolic rate constant (*K*), the coefficient of determination (*r*) and mass gain of Ni-based alloys oxidized at 800 °C for 200 h.

| Alloys | Mass Gain (mg·cm$^{-2}$) | $K$ (mg$^2$·cm$^{-4}$·s$^{-1}$) | $r$ |
|---|---|---|---|
| Pure Ni | 2.9 | $1.17 \times 10^{-5}$ | 0.998 |
| Ni-3Ta | 1.7 | $4.01 \times 10^{-6}$ | 0.999 |
| Ni-6Ta | 3.2 | $1.42 \times 10^{-5}$ | 0.994 |
| Ni-10Ta | 3.8 | $2.01 \times 10^{-5}$ | 0.996 |
| Ni-3Y | 7.7 | $8.24 \times 10^{-5}$ | 0.989 |
| Ni-6Y | 12.2 | $2.06 \times 10^{-4}$ | 0.998 |
| Ni-10Y | 13.8 | $2.65 \times 10^{-4}$ | 0.998 |

Figure 5 presents the X-ray diffraction patterns of Ni-(3~10) Ta alloys and Ni-(3~10) Y alloys oxidized at 800 °C for up to 200 h. The oxidation products of Ni-(3~10) Ta alloys (Figure 5a–c) are NiO and $Ta_2O_5$, showing a similar trend as the oxidation progresses with time. Thus, as with the Ni-(3~10) Y alloys (Figure 5d–f), the oxidation products are NiO and $Y_2O_3$. This proves that the change in Ta or Y content has no effect on the composition of the oxidation product. Comparing Figure 5a with Figure 5b,c, we can find that with the extension of oxidation time, $Ta_2O_5$ is little in the Ni-3Ta alloys but more in the Ni-6Ta and Ni-10Ta alloys. Chicard et al. have reported that when the additional amount of Ta was about 1 wt % in the Ti(C, N)-based cermets, no corresponding oxide was detected during the oxidation, and Ta was mainly doped into the oxide in the form of solid solution phase [16]. In addition, the oxidation process was discovered to be determined by the outward diffusion of Ni ions and inward diffusion of O atoms [24]. Therefore, it can be deduced that Ta could positively enhance

the oxidation resistance of Ni-3Ta alloys by forming Ta$^{5+}$ in oxide layer (NiO) to reduce the oxygen vacancy concentration, and further suppress the inward diffusion of O. However, as for Ni-6Ta and Ni-10Ta alloys (Figure 5b,c), the oxidation products consist of Ta$_2$O$_5$ and NiO after oxidizing for 200 h. The XRD patterns of Ni-(3~10) Y alloys oxidized at 800 °C for up to 200 h are presented in Figure 5d–f, Y is easily oxidized to form Y$_2$O$_3$ in the oxidation layer. After melting, the XRD of the Ni-based alloys with the addition of Y consisted of Ni, Ni$_3$Y, and Ni$_4$Y phase, which is consistent with the EDS analysis in Figure 2.

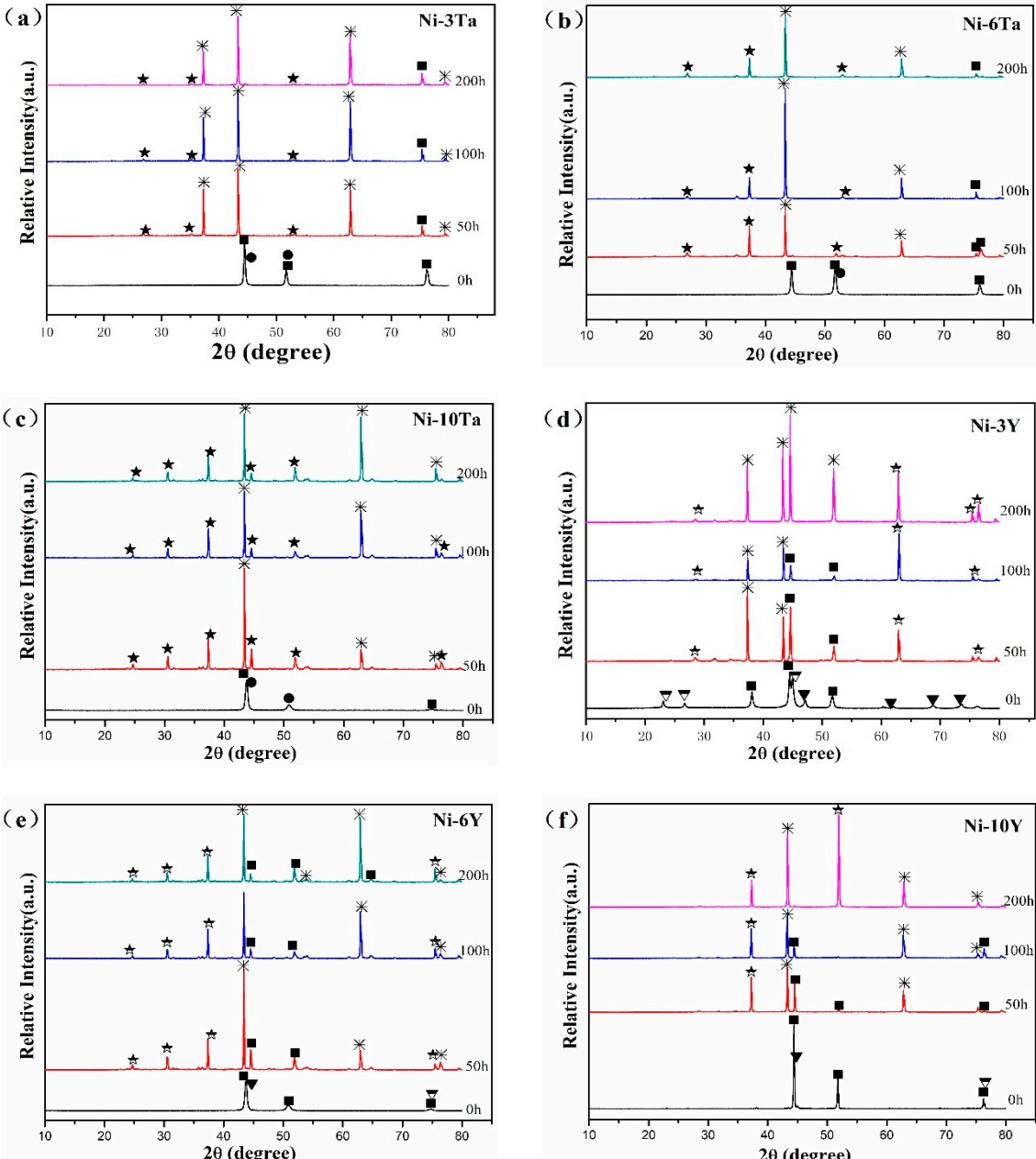

**Figure 5.** XRD patterns of Ni-(3~10) Ta and Ni-(3~10) Y alloys at 800 °C for up to 200 h: (**a**–**c**) Ni-(3~10) alloys, (**d**–**f**) Ni-(3~10) alloys. Phases: (■) Ni; (●) Ni$_2$Ta; (▼) Ni$_4$Y; (▬) Ni$_3$Y; (★) Ta$_2$O$_5$; (▬) Y$_2$O$_3$; (▬) NiO.

The SEM micrographs of samples oxidized in static air at 800 °C for 200 h are presented in Figure 6. For a better analysis of the oxidation process of Ni-(3~10) Ta and Ni-(3~10) Y alloys. The SEM micrographs of pure Ni samples (Figure 6a) is used for comparative analysis. Comparing Figure 6a with Figure 6b–e, the effect of Ta on the oxidation resistance of Ni-based alloys can be summarized as follows. Firstly, the average grain size of surface oxide of pure Ni is greater than of the Ni-3Ta alloys, and close to that of the Ni-6Ta and Ni-10Ta alloys. The low addition of Ta contributes to enhancing the oxidation resistance of Ni-based alloys by refining the size of NiO oxide grain to form more dense oxide layer. Secondly, Figure 6c is a partial topography of the Ni-3Ta alloys (Figure 6b). The surface of the oxide layer is flat and dense, and the grains are uniform, which helps to suppress further oxidation. Lastly, as for the Ni-6Ta and Ni-10Ta alloys (Figure 6d,e), there were some pores on the surface, which were related to the formation of $Ta_2O_5$ oxides. The loose and mixed oxide layer of $Ta_2O_5$ and NiO could not prevent the diffusion of oxygen atoms, but promoted the oxidation process.

As for the Ni-(3~10) Y alloys (Figure 6f–h), the addition of Y had no effect on the surface topography of the oxide layer. As the added amount of Y increased from 3 wt % to 10 wt %, the mass gain of the Ni-(3~10) Y alloys ranged from 7.7 mg·cm$^{-2}$ to 13.8 mg·cm$^{-2}$ (listed in Table 2), all of which were higher than that of the pure Ni. As Y content increased from 3 wt % to 10 wt %, the irregular bonding surface became more and more obvious. According to the EDS and XRD analyses, $Y_2O_3$ mainly concentrated in the junction of the matrix and oxide layer. It can be explained that the addition of Y can strengthen the adhesion between the matrix and the oxide layer by changing the growth mode of the oxide layer on account of the pinning effect of $Y_2O_3$. The adhesion of the substrate and oxide layer was best under the addition of 3 wt % Y in the Ni-based alloys. However, the growth mode changing from plane growth to fibrous growth also increased the contact area of the substrate with air, and then promoted oxidation.

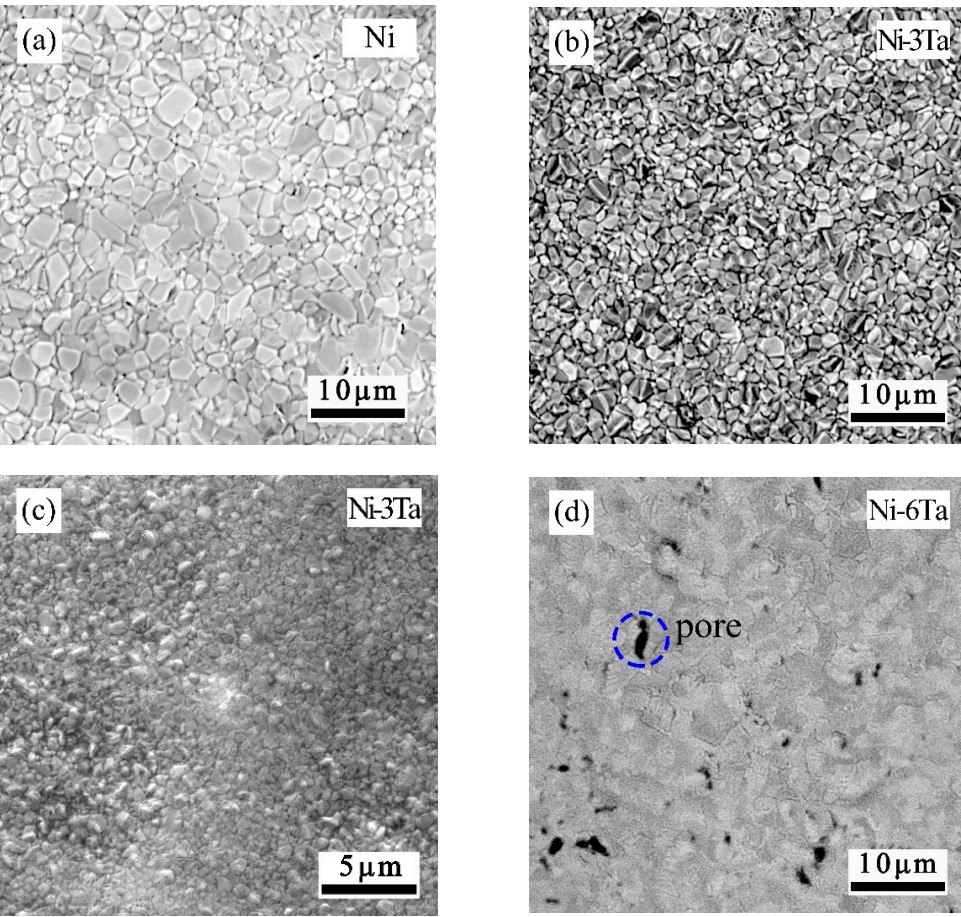

**Figure 6.** *Cont.*

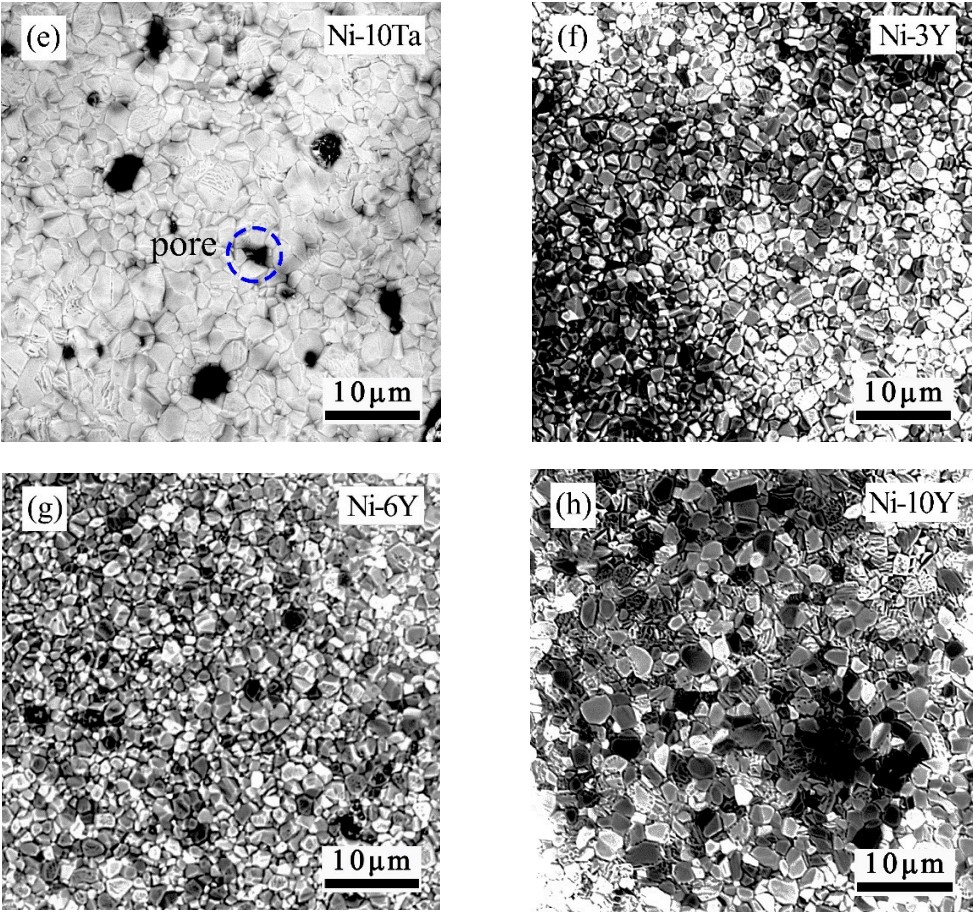

**Figure 6.** SEM micrograph of surface of samples oxidized at 800 °C for 200 h: (**a**) pure Ni, (**b**,**c**) Ni-3Ta, (**d**,**e**) Ni-(6~10) Ta, (**f**–**h**) Ni-(6~10) Y.

## 4. Conclusions

The Ni-(3~10) Ta and Ni-(3~10) Y alloys were fabricated by vacuum arc melting, using argon gas as protective gas, and working electric current of about 220 A. The actual content of Ta or Y in the Ni-based alloys was confirmed to be close to the theoretical content by using X-ray fluorescence analysis. According to the XRD patterns and SEM images of Ni-Me (Me = Ta or Y) alloys after melting, the doping of Ta and Y was uniform, and fully melted without individual Ta or Y phase. After the cyclic and isothermal oxidation at 800 °C for 200 h, the Ni-based alloys with 3 wt % addition of Ta presented better oxidation resistance by forming the $Ta^{5+}$ ions to restrict the inward diffusion of oxygen. The addition of Y decreased the oxidation resistance. However, it was worth noting that the pinning effect of $Y_2O_3$ could heighten the adhesion between the matrix and the oxide layer by changing the growth pattern. Through the experimental results, we can know the specific effect of Ta or Y element on the high-temperature oxidation resistance of Ni-based alloys, which will be helpful for us in trying to improve the oxidation resistance of cermet in future research. The investigated alloys are the potential candidates for the matrix of TiN/Ni-Me cermet material, instead of pure Ni. The results obtained within the present work show, that almost all alloys show worse oxidation behavior than pure Ni. It seems to be that only Ni-3Ta can be potentially considered as a candidate for the cermet matrix. We also know that there are some deficiencies in the experiments; we need to design the Ni-based alloys with integrated addition of Ta and Y for comparative experiment. However, it is hard to melt the Ni-Ta-Y ternary alloys by vacuum arc melting. To overcome this issue, the TiN/Ni-3Ta cermet with minor $Y_2O_3$ addition can be fabricated by spark plasma sintering. Moreover, the novel cermet could

combine the excellent effect of Ta and Y to enhance the high-temperature oxidation resistance, and then be used as SOFC interconnects with better properties, which will be the focus of our future research.

**Author Contributions:** H.D. performed the experiments and wrote the paper; Y.L. and T.L. analyzed the data; H.Z. searched the reference; Z.H. performed the microstructural analysis and edited the paper.

**Funding:** The study received financial support from the National Natural Science Foundation of China (Grant No. 51671209).

**Acknowledgments:** The authors are grateful for the financial support from the National Natural Science Foundation of China (Grant No. 51671209).

**Conflicts of Interest:** The authors declare no conflict of interest.

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
