# Peer review of "Investigation on the High-Temperature Oxidation Resistance of Ni-(3~10) Ta and Ni-(3~10) Y Alloys"

_metals, doi:10.3390/met9010097_

Reviewer 1 Report

The manuscript presents many flaws that make it not suitable for publishing. In the following points are the reasons motivating rejection.

1) The title of the manuscript is confusing. It seems that the study deals with the oxidation behavior of ternary Ni-Ta-Y alloys and not different binary Ni-Ta or Ni-y alloys.

2) XRD patterns of Ni-Ta and Ni-Y alloys are almost identical. No peaks due to intermetallic Ni-Ta or Ni-Y are identified in spite microstructural observation evidence significant volume fractions of these intermetallic phases. Only for the Ni-3Y alloy, peaks attributed to the intermetallic phases are clear, but surprisingly no peaks are view for the Ni-10Y when the volume fraction of intermetallic compounds is much higher.

3) Labelling of Fig. 2 is wrong. Predominant phase is the nickel matrix (dark phase). Labels should be properly marked in the micrographs.

4) Fig. 3b is wrong (It is the same than that of Fig.4)

5) In figure 5 is absent the symbol of NiO oxide in the figure caption. Identification of the peaks should be carefully revised. For instance, one of the two peaks identified as NiO (2theta about 45°) cannot be attributed to NiO in Fig. 5d.

6) Line 130, replace Ta2O5 by Ta2O3

7) Line 150, replace Ni3Y and Ni4Y by Ni3y and N4Y

8) Quality of cross sectional views in Fig. 7 is extremely poor. No information can be extracted from these images. Samples are badly polished so no information can be obtained from the matrix and the oxide scale. Thus, there is no information about the influence of second phases on the oxidation behaviour of the alloy. Furthermore, calculated thicknesses of the oxide scale are wrong because the surface is not smooth and part of the measure comes from the edges. Finally, these images do not provide information about the structure and composition of the scale. No microanalytical results are presented. All this data are required for an appropriate discussion about the oxidation behaviour of these alloys.

9) The choice of expensive elements such Y and Ta as alloying additions is difficult to understand. Al and/or Cr has proven to be very effective for improving mechanical and oxidation behaviour of nickel based alloys. What could be the advantages of using Y and Ta instead of these elements in the applications mentioned in the introduction? This should be properly addressed in the introduction section.

10) The beneficial effect of Ta additions is related to the fact that this element dissolves in the NiO lattice reducing the oxygen vacancy in NiO formed in Ni-3Ta alloy. Why cannot the same argument applied for the Ni-6Ta or Ni-10Ta alloys? Ta dissolution in NiO lattice could be compatible with formation of Ta2O5 when Ta content increases in the alloy. Moreover, the authors have not considered that oxygen transport can proceed along grain boundaries of the oxide scale, so major effect of such elements could be related to blocking/delaying diffusion of elements building up the scale, as found in chromia-or alumina-former alloys.

Author Response

Point 1: 1) The title of the manuscript is confusing. It seems that the study deals with the oxidation behavior of ternary Ni-Ta-Y alloys and not different binary Ni-Ta or Ni-y alloys.

Response 1: Thanks for your advice. I have changed the title. “Investigation on the high temperature oxidation resistance of Ni-(3~10) Ta and Ni-(3~10) Y alloys.” is the new title.

Point 2: XRD patterns of Ni-Ta and Ni-Y alloys are almost identical. No peaks due to intermetallic Ni-Ta or Ni-Y are identified in spite microstructural observation evidence significant volume fractions of these intermetallic phases. Only for the Ni-3Y alloy, peaks attributed to the intermetallic phases are clear, but surprisingly no peaks are view for the Ni-10Y when the volume fraction of intermetallic compounds is much higher.

Response 2: Thanks for your suggestion. It is my fault that one of the dots is missing in the Ni-10Y. And I have added it. The XRD patterns of Ni-Ta are really different from the Ni-Y alloys. There are NiXMey phase peaks in the XRD patterns. The Ni-10 Y alloys still have the Ni-Y solid solution phase peaks, which are obvious.

Point 3: 3) Labelling of Fig. 2 is wrong. Predominant phase is the nickel matrix (dark phase). Labels should be properly marked in the micrographs.

Response 3: Thanks very much for your kind advice. It is my fault to falsely mark the labels in the micrographs. I have corrected them.

Point 4: 4) Fig. 3b is wrong (It is the same than that of Fig.4)

Response 4: Thanks for your kind reminders. I am so sorry to put a wrong picture in the Fig.3. I have checked it and replaced it with the parabolic Figure.

Point 5: 5) In figure 5 is absent the symbol of NiO oxide in the figure caption. Identification of the peaks should be carefully revised. For instance, one of the two peaks identified as NiO (2theta about 45°) cannot be attributed to NiO in Fig. 5d.

Response 5: Thanks for your kind reminders. I have added the symbol of NiO oxide “”NiO in the Fig. 5 caption. And I have checked the identification of the peaks throughout all Figures carefully again.

Point 6: 6) Line 130, replace Ta2O5 by Ta2O3

Response 6: The stable oxide of Ta element is Ta2O5. According to the XRD, the Ta element form Ta2O5 oxide. Maybe the reviewer needs me to replace the Ta2O3 with Ta2O5.

Point 7: 7) Line 150, replace Ni3Y and Ni4Y by Ni3y and N4Y

Response 7: Thanks for your kind advice. I have replaced them.

Point 8: Quality of cross sectional views in Fig. 7 is extremely poor. No information can be extracted from these images. Samples are badly polished so no information can be obtained from the matrix and the oxide scale. Thus, there is no information about the influence of second phases on the oxidation behaviour of the alloy. Furthermore, calculated thicknesses of the oxide scale are wrong because the surface is not smooth and part of the measure comes from the edges. Finally, these images do not provide information about the structure and composition of the scale. No microanalytical results are presented. All this data are required for an appropriate discussion about the oxidation behaviour of these alloys.

Response 8: Thanks for your analysis. Before oxidation test, the samples have been strictly polished by several process. I think the surface roughness of the samples can be able to meet the requirements of the experiment. The thickness is calibrated as close as possible to the actual. Your advice is also very helpful to me. In the future research, I will focus more on the characterization of the samples, and make the picture beautiful for reader to better understand.

Point 9: 9) The choice of expensive elements such Y and Ta as alloying additions is difficult to understand. Al and/or Cr has proven to be very effective for improving mechanical and oxidation behavior of nickel-based alloys. What could be the advantages of using Y and Ta instead of these elements in the applications mentioned in the introduction? This should be properly addressed in the introduction section.

Response 9: Thanks for your kind suggestion. I have added this discussion in the first paragraph of introduction section. In this paper, the goal of the investigation of Ni-Ta and Ni-Y alloys is to choose the alloys with best oxidation resistance as the matrix of the TiN/Ni-Me cermets applied for SOFC interconnects. The Cr and Al are known to improve the oxidation resistance. But Cr can form Cr2O3 to deposit on the cathode and degrade the SOFC performance. And the high temperature conductivity of Al2O3 is extremely poor, which will reduce the electrical conductivity. Thus, these two elements should not be chosen. We need to find new metal element to enhance oxidation resistance. According to the comprehensive analysis, the investigation of Ni-Ta and Ni-Y alloys is necessary.

Point 10: The beneficial effect of Ta additions is related to the fact that this element dissolves in the NiO lattice reducing the oxygen vacancy in NiO formed in Ni-3Ta alloy. Why cannot the same argument applied for the Ni-6Ta or Ni-10Ta alloys? Ta dissolution in NiO lattice could be compatible with formation of Ta2O5 when Ta content increases in the alloy. Moreover, the authors have not considered that oxygen transport can proceed along grain boundaries of the oxide scale, so major effect of such elements could be related to blocking/delaying diffusion of elements building up the scale, as found in chromia-or alumina-former alloys.

Response 10: The reviewer’s analysis is very helpful to me, which brings me many inspirations. From the analysis of the mass gain of Ni-Ta alloys with different addition. The mass gain of pure Ni is higher than of theNi-3Ta alloys, but lower than of the Ni-6Ta and Ni-10Ta alloys. Thus, the low addition content of Ta can improve the oxidation resistance. As suggested by the reviewer, the oxygen transport maybe proceed along grain boundaries of the oxide scale. This diffusion approach has given me a lot of inspiration. I think my existing data is enough to characterize the oxidation resistance of Ni-Me alloys. But I will do more work on in-depth characterization of the oxidation resistance in the future, according to the reviewer’s advice.

Reviewer 2 Report

The high temperature oxidation resistance of Ni alloys is truly important to many parts of the energy conversion industry. Ni alloys tend to have higher strength and oxidation resistance than Fe alloys at high temperatures and finding additions that will further improve their oxidation/corrosion resistance is an important endeavor. Reading this paper gave me additional understanding on the use of Ta or Y to help limit the oxidation.

Based on the data presented the results seem justified.

See attached for further comments

Author Response

I am very grateful to the reviewer for patiently reviewing my article. Your advice is very helpful for me to modify the manuscript.

Point 1: There are areas of……would be meaningful as well.

Response 1: Thanks for your kind suggestion. The size of the melting pool is Ф 40mm * 15mm, which is big for melting. In the actual operation process, the weight of samples is 45g per each time. I did it as the reviewer said. The samples are melting several times to improve the homogeneity. But for the SEM images of Ni-based alloys, I have done the EDS analysis several time, the results are consistent. I have checked the binary phase diagram of Ni-Ta alloys and Ni-Y alloys. The melting point of Ni2Ta, Ni4Y, and Ni3Y is 1405°C, 1375 °C, and 1237 °C, respectively. The reviewer’s analysis is correct. After receiving the review comments, I did the DSC analysis again. The original data was really fault. I have rewritten this part in the resubmit manuscript.

Point 2: Your Fig 2 should say which specific alloy is being shown in the image (3, 6, or 10 wt. % secondary element.)

Response 2: Thanks for your kind reminders. I have rewritten the caption of the Fig. 2

Point 3: (2.1 Sample Prep) As well as give the specimen size it would be nice to know the approximate specimen weight. How were the specimen supported in the furnace, to allow air to get to all the surfaces? Did you determine the grain side of the melts. Does the addition of Ta result in a finer grain oxide (Fig 6 and line 161) which has a improved oxidation resistance, or does the addition of Ta result in a finer grain bulk structure, which results in finer grain oxides and improved oxidation resistance?

Response 3: Thanks for your suggestion. The weight of sample for melting is not specific. This is relevant to the size of melting pool of the vacuum arc melting. In my experiment, the weight for every melting is 45g. I think it is not significant, so I have not written it in the manuscript. In the oxidation process, the specimen was put on the Al2O3 ceramics flake to allow air to get all the surfaces. The grain size of the pure Ni is about 4-7 µm, and the grain size of the Ni-3Ta is 2-3µm. From the fig. 6(a) and fig. 6(c), we are very clear to see the difference in size between the two figures. Comparing the SEM micrograph of the Ni-3 Ta alloys with other images, the low content of Ta element can improve the oxidation resistance.

Point 4: (2.2 Oxidation and characterization) Why did you chose 800 C for your oxidation tests, rather than, say, 600 C? These seems to be the upper end for SOFC operations. Can you expand on this a little? Did you put your samples in a hot furnace or did you start with a cold furnace and heat the samples? If you put them in a cold furnace, how long did it take to get to temperature. This could be significant with your fairly short exposure times.

Response 4: The working temperature of SOFC is 600-800 °C. The reason of that I chose 800 °C for my oxidation tests is that I want to choose the alloys with best oxidation resistance. If the alloy has good oxidation resistance at 800 °C, it will work better at 600 °C. So, 800 °C is more representative. I put the samples in a cold furnace and heat the samples to make the sample evenly heated. It takes about 40 minutes to get to temperature. The heating time is negligible to the oxidation time.

Point 5: (3.2 High temperature oxidation resistance of the Ni-based alloys) Are these the results of single samples or did you use multiple samples of each composition?

Response 5: I have prepared several samples for each composition in the experiment in order to get more convincing results. Like the mass gain, it is the average value of multiple samples of each composition. Thus, the value of mass gain will be more accurate.

Point 6: No scale bar for Fig 7 f, should we assume it is the same as Fig 7 e?

Response 6: I have added the scalebar in the Fig. 7f.

Point 7: There are multiple English spelling and grammer issues, I have attached a mark on copy of your manuscript with some suggestions. Some specific suggestions follow

Response 7: Thanks for your nice suggestion. I have checked all of them and rewritten them.

Point 8: A comment concerning your problems preparing ternary alloys. You could prepare two binary alloys, one of Ni-Ta and one of Ni-Y and then combine the two binaries in chosen ratios to make your ternary alloy.

Response 8: Thanks for yours good advice. It is very helpful to me, which gives me a new inspiration. I will try it in the further experiment.

Point 9: commented [A1] the added diversity of elements

Response 9: I have replaced this part with “the diversity of added elements”.

Point 10: Commented [A3]: Just dendrites, if you could see the structure.

Response 10: Thanks for your kind reminders. I have rewritten it.

Reviewer 3 Report

The paper „Investigation on the High Temperature Oxidation Resistance of Ni-(3-10) Ta-(3-10) Y Alloys” describes the influence of alloying elements on oxidation behavior of pure nickel. The Authors found that only addition of 3 wt. % of Ta causes lowering of the oxidation rate of Ni, while the addition of 6 and 10 wt. % increase the oxidation rate. For all alloys containing Y-additions (3, 6 and 10 wt.%) substantial increase of Ni oxidation kinetics was observed. Te Authors claimed, that the investigated alloys could potentially be used as an interconnectors in SOFC stacks.

During the reading a several questions/comments arises, as following:

1.       Page 1, lines 19 and 20: In the text: “2.9 mg·cm-2 to 1.7 mg·cm-2” is written, while should be:  “2.9 mg·cm-2 to 1.7 mg·cm-2”,

2.       Page 2, line 50: In the text: “Ta5+” is given, while 5+ should be a superscript: Ta5+ ,

3.       Page 2, line 70: The Authors should not start the sentences from “And”. Please re-write such sentences.

4.       Page 3, Figure 2:
a) The scale-bar is missing for both images. Please add them.
b) It is not clear which mode of SEM the Authors used? SE or BSE? Normally, heavier elements gives brighter signals in SEM images due to their larger number of electrons which can interact with electron beam. In the case of images in Figure 2 a and b, brighter regions are identified to be pure Ni. Are the Authors sure about this? Please comment on this point.
c) The Figure 2 caption says, that in Figure “SEM images of the Ni-based alloys: (a) Ni-(3-10) Ta, (b) Ni-(3-10) Y” are shown, while only two images showing the microstructure of the alloys in as-cast condition is shown. Moreover, the reader do not know which alloy chemistry is shown in certain figure. Please specify, morphology of which alloy chemistry is depicted in Figure 2 a and 2 b.

5.       Page 4, Figure 3: In the Figure 3 caption, the Authors stated that the mass-gains are shown, while Figure 3 b depicts SQUARED MASS GAIN (exactly the same as Figure 4 b). Therefore, please correct the image shown in Figure 3 b.

6.       Page 5, line 138: Tha Authors wrote, that “Ta2O3” is formed in Ni-3Ta alloy, while in the analytical part the Authors stated formation of Ta2O5. Please clarify this.

7.       Page 5, line 150: The Authors wrote in the text that: “..Y addition consist of Ni, N3Y and N4Y phase..” Are the Authors sure that Y formed N3Y and N4Y nitrides? Please show the results confirming this findings.

8.       Page 5 and 6, Figure 5: Please arrange in similar way as in Figure 1, in term of showing the legend within the image, not in the Figure caption. Moreover, in Figure caption, the Authors also stated that they found N4Y phase. Please check this finding once again.

9.       Page 6, lines 178-179: The Authors concluded, that the addition of Y improves adhesion of the oxide scale to the matrix. This is well known effect and described previously by e.g. Gil et al. Surface and Coatings 201 (2006), 3824-3828.

10.    Page 7, Figure 6: The SEM images of the surfaces after oxidation presented in Figure 6 do not bring to much information. Moreover, it is not clear why the alloy Ni3-Ta is shown twice (Figure 6b and c) in different magnification? Please add also information about the SEM mode used (SE or BSE??). The quality of the images is quite poor. Additionally, no quantitative measurement were done (or they were not shown in the present work), therefore, only information about the oxide scale thicknesses are shown.

11.    Page 9, Chapter Conclusions: In the present form it is rather short summary of finding that conclusions. Moreover, the effect of Y2O3 in cermet can be different than for pure metallic alloy and it was not investigated in the present work, therefore comment about this materials is not necessary here.

12.    Page 9, Authors Contributions: The Authors contribution is missing.

13.    Page 9 and 10, References: Most of the references are older than 10 years. For example, the Authors discuss the findings on the effect of Ti and Ta and cite the work of Yang from 1981, while nowadays the effects of these elements were also investigated, e.g. by Jalowicka et al. Materials and Corrosion 65, 2014, 178-187 or Nowak et al. High Temperature Materials and Processes 37(9-10), 2018, pp. 801-806.

14.    General comment: Why the Authors investigated the model alloys containing alloying elements varied between 3 and 10 wt.% ? Especially interesting is addition of Y in the range between 3 and 10 wt. %, because in the literature it is known that something like “overdoping effect” exist, especially for rare earth elements (like Y). The overdoping effect was previously described by e.g. Klöver, Materials and Corrosion 51, 2000, 373-385.

15.    General comment: Investigated alloys are the potential candidates for the matrix of TiN/Ni-Me cermet material, instead of pure Ni. The results obtained within the present work, show, that almost all alloys show worse oxidation behavior than pure Ni. It seems to be, that only Ni-3Ta can be potentially considered as a candidate for cermet matrix.

16.    General comment: The interconnectors in SOFC works in two types of the atmospheres: high pO2 in cathode side and low pO2 in the anode side. The second usually is simulated using a gas mixture consisting of Ar-4%H2-10%H2O. How the selected materials would behave in the anode gases? Did the Authors conducted the tests in such low pO2 gases? How to transfer the data obtained in the present work to the real condition in SOFC?

17.    General comment: Moderate English corrections are demanded.

Author Response

Point 1: Page 1, lines 19 and 20: In the text: “2.9 mg·cm-2 to 1.7 mg·cm-2” is written, while should be:  “2.9 mg·cm-2 to 1.7 mg·cm-2

Response 1: Thanks for your kind reminders. I have replaced it.

Point 2: Page 2, line 50: In the text: “Ta5+” is given, while 5+ should be a superscript: Ta5+

Response 2: Thanks for your kind reminders. This mistake may be produced in the process that the editorial office made formatting changes to my original submission. I have replaced it.

Point 3:        Page 3, Figure 2:
a) The scale-bar is missing for both images. Please add them.
b) It is not clear which mode of SEM the Authors used? SE or BSE? Normally, heavier elements gives brighter signals in SEM images due to their larger number of electrons which can interact with electron beam. In the case of images in Figure 2 a and b, brighter regions are identified to be pure Ni. Are the Authors sure about this? Please comment on this point.
c) The Figure 2 caption says, that in Figure “SEM images of the Ni-based alloys: (a) Ni-(3-10) Ta, (b) Ni-(3-10) Y” are shown, while only two images showing the microstructure of the alloys in as-cast condition is shown. Moreover, the reader do not know which alloy chemistry is shown in certain figure. Please specify, morphology of which alloy chemistry is depicted in Figure 2 a and 2 b.

Response 4: Thanks for your advice. I have added the scalebar. I used BSD mode to analyze the phase. It is my fault to falsely mark the labels in the micrographs. I have corrected them. Predominant phase is the nickel matrix (dark phase). In the Fig. 2, the Fig. 2a is the Ni-3 Ta alloys and the Fig. 2b is the Ni-3 Y alloys. I have added it and rewritten the sentence.

Point 5: Page 4, Figure 3: In the Figure 3 caption, the Authors stated that the mass-gains are shown, while Figure 3 b depicts SQUARED MASS GAIN (exactly the same as Figure 4 b). Therefore, please correct the image shown in Figure 3 b.

Response 5: Thanks for your kind reminders. I am so sorry to put a wrong picture in the Fig.3. I have checked it and replaced it with the parabolic Figure.

Point 6: Page 5, line 138: That Authors wrote, that “Ta2O3” is formed in Ni-3Ta alloy, while in the analytical part the Authors stated formation of Ta2O5. Please clarify this.

Response 6: Thanks for your kind reminders. It is the spelling mistake. Line 138, I have replaced the Ta2O3 with Ta2O5.

Point 7: Page 5, line 150: The Authors wrote in the text that: “..Y addition consist of Ni, N3Y and N4Y phase..” Are the Authors sure that Y formed N3Y and N4Y nitrides? Please show the results confirming these findings.

Response 7: It is the spelling mistake. I have replaced the N3Y and N4Y with Ni3Y and Ni4Y, respectively.

Point 8: Page 5 and 6, Figure 5: Please arrange in similar way as in Figure 1, in term of showing the legend within the image, not in the Figure caption. Moreover, in Figure caption, the Authors also stated that they found N4Y phase. Please check this finding once again.

Response 8: Thanks for your kind advice. But in the Figure 5, too many phases need to be marked. If I put them in the image, the image will look crowded and untidy. This marked method has been applied in many journals. The N4Y is the spelling mistake. I have replaced it with Ni4Y.

Point 9: Page 6, lines 178-179: The Authors concluded, that the addition of Y improves adhesion of the oxide scale to the matrix. This is well known effect and described previously by e.g. Gil et al. Surface and Coatings 201 (2006), 3824-3828.

Response 9: Thank you very much for recommending this article to me. “Effect of surface condition on the oxidation behavior of MCrAlY coatings”. I have read it carefully. In this paper, the authors focus on the effect of different surface roughness on the oxidation behavior of multiple alloys coatings. But it is not easy and accurate to analyze the specific effect of the Y element due to the diversity of elements. Most of the time, the paper discusses the coaction mechanism of Y and Al. The goal of my research is to choose the alloys with best oxidation resistance as the matrix of the TiN/Ni-Me cermets applied for SOFC interconnects. But Al is not the good choose in our research, which I have discussed in the introduction section. In my manuscript, different method has been used to characterize the effect of Y and more benefit for reader to understanding the specific effect of Y. The investigation on the oxidation resistance of Ni-Y alloys is significative.

Point 10: Page 7, Figure 6: The SEM images of the surfaces after oxidation presented in Figure 6 do not bring to much information. Moreover, it is not clear why the alloy Ni3-Ta is shown twice (Figure 6b and c) in different magnification? Please add also information about the SEM mode used (SE or BSE??). The quality of the images is quite poor. Additionally, no quantitative measurement were done (or they were not shown in the present work), therefore, only information about the oxide scale thicknesses are shown.

Response 10: Thanks for your suggestion. Because from the experiment analysis, the Ni-3 Ta alloys have good oxidation resistance. So, I need to focus more on the microstructure. In the Figure 6, the Ni-3 Ta alloys is shown twice in different magnification in order to make the microstructure clearer for reader. The SEM mode used BSD. I have done my best to provide the good quality of the images. Maybe the color contrast is not very well. Thanks again for your good advice. I will improve my characterization skill in the future research. Your suggestion is very helpful to me.

Point 11: Page 9, Chapter Conclusions: In the present form it is rather short summary of finding that conclusions. Moreover, the effect of Y2O3 in cermet can be different than for pure metallic alloy and it was not investigated in the present work, therefore comment about this materials is not necessary here.

Response 11: In my manuscript, I put the results and discussions together. So, the conclusions may look a bit short. Most of my data analysis is in the “results and discussions” section. I can guarantee that I have given a complete manuscript, which can display my previous research work. I known the Y2O3 is a sintering aid in the ceramic sintering process in the most of time. But the cermet sintering process always don’t need sintering aid. Maybe a minor content Y2O3 has specific effect on the cermet sintering process with the Ta addition. This is just my guess. Bold speculation may have greater innovation.

Point 12: Page 9, Authors Contributions: The Authors contribution is missing.

Response 12: Thanks for your kind reminders. I have added it.

Point 13: Page 9 and 10, References: Most of the references are older than 10 years. For example, the Authors discuss the findings on the effect of Ti and Ta and cite the work of Yang from 1981, while nowadays the effects of these elements were also investigated, e.g. by Jalowicka et al. Materials and Corrosion 65, 2014, 178-187 or Nowak et al. High Temperature Materials and Processes 37(9-10), 2018, pp. 801-806.

Response 13: Thanks for your advice. I have read the two articles carefully. The article “Effect of nickel base superalloy composition on oxidation resistance in SO2 containing, high pO2 environments” is very helpful to me. I have quoted it. In my references, references after 2014 occupy twelve. Reference between 2008 and 2014 occupy six. Just little references are older than 10 years. Moreover, your suggestion is helpful to me. I have replaced some old reference again.

Point 14: General comment: Why the Authors investigated the model alloys containing alloying elements varied between 3 and 10 wt.% ? Especially interesting is addition of Y in the range between 3 and 10 wt. %, because in the literature it is known that something like “overdoping effect” exist, especially for rare earth elements (like Y). The overdoping effect was previously described by e.g. Klöver, Materials and Corrosion 51, 2000, 373-385.

Response 14: Thanks for you recommending the article “factors affecting the oxidation behavior of thin Fe-Cr-Al foils part II: The effect of alloying elements: overdoping”. In this article, the oxidation behaviour of iron-chromium-aluminium alloys containing 7 mass-% aluminium, additions of silicon and reactive elements (Hf, Y, Ti, Ce, La, Zr) was investigated at temperatures ranging between 900 and 1300 °C. Firstly, the article involves the interaction of multiple elements, the conclusion may not apply to the condition of a single element addition. Secondly, the conditions of oxidation are also different, the “overdoping effect” may not always exist in all materials. However, the reviewer’s advice is very helpful to me. I have learned a lot from this article. In the future research, I will further optimize my experiment plan.

Point 15: General comment: Investigated alloys are the potential candidates for the matrix of TiN/Ni-Me cermet material, instead of pure Ni. The results obtained within the present work, show, that almost all alloys show worse oxidation behavior than pure Ni. It seems to be, that only Ni-3Ta can be potentially considered as a candidate for cermet matrix.

Response 15: Thanks for wonderful analysis.

Point 16: General comment: The interconnectors in SOFC works in two types of the atmospheres: high pO2 in cathode side and low pO2 in the anode side. The second usually is simulated using a gas mixture consisting of Ar-4%H2-10%H2O. How the selected materials would behave in the anode gases? Did the Authors conducted the tests in such low pO2 gases? How to transfer the data obtained in the present work to the real condition in SOFC?

Response 16: As the reviewer said, the atmosphere of the cathode is high pO2 and the atmosphere of the anode is mixed gas of Ar-4%H2-10% H2O. The goal of the investigation of Ni-Ta and Ni-Y alloys is to choose the alloys with best oxidation resistance as the matrix of the TiN/Ni-Me cermets applied for SOFC interconnects. The effect of high pO2 on the Ni-based alloys is higher than of mixed gas, according to our previous work. In this manuscript, the investigation of Ni-based alloys is the first step. Next, I will focus on the oxidation resistance of TiN/Ni-Me cermets under cathode and anode environment of SOFC. The research on the oxidation resistance in the low pO2 gas will be done in the future research.

Point 17: General comment: Moderate English corrections are demanded

Response 17: I know there are some grammar and English spelling issues in the manuscript. I have checked the whole manuscript and done my best to improve the quality of the resubmit manuscript.

Round  2

Reviewer 1 Report

The authors have introduced changes in the manuscript, but the quality of thit is not enough for publishing. Discussion is mainly not supported by experimental data. The authors state that beneficial effect of 3% Ta addition comes from Ta5+ dissolved in the NiO scale. This could be true but the same effect could be expected for Ni-6La and Ni-10La alloys. In addition, no EDS data are provided to check such Ta dissolution in the NiO scale. Moreover, the accuracy of XRD identification is weak. The same peaks are ascribed to different phases depending on the composition of the alloys. Such inconsistencies cannot be present in XRD patterns.

From cross sectional micrographs no information about the oxide scale can be obtained. Quality of the images is extremely bad. They should be changed by new images in which statements made by the authors could be seen. This reviewer recommends the deposit of nickel or copper over the oxide scale to prevent cracking and/or losses of the scale during metallographic preparation. From the images provided (from Fig 7a to Fig. 7e) the thickness of the scale cannot be measured. Furthermore, no information about the influence of intermetallic phases existing in the alloy over the structure of the oxide scale can be deduced from the view of Figures 7f, 7g or 7h.

Author Response

Point 1: The authors state that beneficial effect of 3% Ta addition comes from Ta5+ dissolved in the NiO scale. This could be true but the same effect could be expected for Ni-6Ta and Ni-10Ta alloys. In addition, no EDS data are provided to check such Ta dissolution in the NiO scale. Moreover, the accuracy of XRD identification is weak. The same peaks are ascribed to different phases depending on the composition of the alloys. Such inconsistencies cannot be present in XRD patterns.

Response 1: Thanks for your suggestion. From the experiment results, the mass gain of Ni-Ta alloys with different addition are listed in the table 2. The mass gain of pure Ni is higher than of the Ni-3Ta alloys, but lower than of the Ni-6 Ta and Ni-10 Ta alloys. Thus, the low addition of Ta can improve the oxidation resistance. The same effect is not expected for Ni-6Ta and Ni-10 Ta alloys. From the analysis of XRD, the diffraction peak intensity of Ta2O5 in the Ni-6Ta and Ni-10Ta is higher than Ni-3Ta, which is not benefit to improve the high temperature oxidation resistance. It is not right that the same peaks are ascribed to different phases. Smaller pictures result in very small peak intervals, which is related to the layout of the manuscript image. I have been very careful to calibrate the composition of the peaks. Thanks for your kind advice again, in my future work, I will pay more attention to these details.

Point 2: From cross sectional micrographs no information about the oxide scale can be obtained. Quality of the images is extremely bad. They should be changed by new images in which statements made by the authors could be seen. This reviewer recommends the deposit of nickel or copper over the oxide scale to prevent cracking and/or losses of the scale during metallographic preparation. From the images provided (from Fig 7a to Fig. 7e) the thickness of the scale cannot be measured. Furthermore, no information about the influence of intermetallic phases existing in the alloy over the structure of the oxide scale can be deduced from the view of Figures 7f, 7g or 7h.

Response 2: Thanks for your good advice. It is a very good idea that the deposit of nickel or copper over the oxide scale to prevent cracking and/or losses of the scale during metallographic preparation. I will try it in the further work. I have tried my best to provide beautiful pictures. The surface and cross-section micrograph of the oxide scale can be obviously seen in Figure 6 and Figure 7. From Fig.7a to Fig. 7d, the thickness of the oxide scale of Ni and Ni-(3~10) Ta is 18 µm, 13 µm, 21µm, and 38 µm, respectively. From Fig. 7e to Fig. 7h, the thickness of the oxide scale of Ni-(3~10) Y is 50 µm, 62 µm, and 78 µm. The composition of the oxide products of the Ni-(3~10) Ta and Ni-(3~10) Y can be analyzed from the XRD (Figure 5). Figure 7 (a-e) shows the different structure and micrograph of the Ni-(3~10) Ta and Ni-(3~10) Y. From Fig. 7, we can obviously observe that the different alloys (Ta and Y) and different content have a different effect on the cross section of the oxide layer of the Ni-based alloys. I think that the experiment data is enough to support my conclusion. I also know that my current work is not good enough, and the reviewer’s advice is also very helpful. I will continue to optimize the characterization of materials in my future work. 

Reviewer 3 Report

The Authors seriously reacted on almost all of the comments and made a necessary corrections. However, one of the comments is neglected by the Authors, namely 

Point 15: General comment: Investigated alloys are the potential candidates for the matrix of TiN/Ni-Me cermet material, instead of pure Ni. The results obtained within the present work, show, that almost all alloys show worse oxidation behavior than pure Ni. It seems to be, that only Ni-3Ta can be potentially considered as a candidate for cermet matrix.

Response 15: Thanks for wonderful analysis.

It was not a purpose of the Reviewer to hear "thank you", but to include such summary clearly in the conclusions, because this is the main message comming from the paper.

I think that after this minor correction, the article can be published in "Metals".

I wish the Authors all the best in future work with this interesting topic.

Author Response

Point 15: General comment: Investigated alloys are the potential candidates for the matrix of TiN/Ni-Me cermet material, instead of pure Ni. The results obtained within the present work, show, that almost all alloys show worse oxidation behavior than pure Ni. It seems to be, that only Ni-3Ta can be potentially considered as a candidate for cermet matrix.

Response 15: Thanks for wonderful analysis.

It was not a purpose of the Reviewer to hear "thank you", but to include such summary clearly in the conclusions, because this is the main message comming from the paper.

I think that after this minor correction, the article can be published in "Metals".

Response: Thanks for the reviewer's good advice, I have included this part in the conclusion in the resubmitted manuscript.

Round  3

Reviewer 1 Report

Good quality of cross sectional images is necessary to get useful information about the oxidation mechanism. Such information cannot be deduced from the provided images. Moreover, the accuracy of thickness scale is very low from these micrographs so such values should be removed from the text unless new images with much higher quality could be included in the text. Since the discussion of the results is not affected by information arising from cross sectional views, the manuscript could be accepted if cross sectional views and thickness values are removed in a new version of the manuscript.

Author Response

Point 1: Good quality of cross sectional images is necessary to get useful information about the oxidation mechanism. Such information cannot be deduced from the provided images. Moreover, the accuracy of thickness scale is very low from these micrographs so such values should be removed from the text unless new images with much higher quality could be included in the text. Since the discussion of the results is not affected by information arising from cross sectional views, the manuscript could be accepted if cross sectional views and thickness values are removed in a new version of the manuscript.

Response 1: Thanks for the reviewer’s good advice. I have removed the Fig. 7 in the resubmitted manuscript,and rewritten this part. Your suggestion is very helpful for me. Thank you again.

Round  4

Reviewer 1 Report

The manuscript is ready for publishing but the following sentence (But comparing the SEM micrograph of the cross-section of samples oxidized at 800 800 °C for 200 h, the bonding surface of oxide layer and matrix is smooth and neat, however the bonding surface is zigzag in the Ni- (3~10) Y alloys) must be removed because the figure regarding cross section has been removed in the new manuscript

Author Response

The manuscript is ready for publishing but the following sentence (But comparing the SEM micrograph of the cross-section of samples oxidized at 800 800 °C for 200 h, the bonding surface of oxide layer and matrix is smooth and neat, however the bonding surface is zigzag in the Ni- (3~10) Y alloys) must be removed because the figure regarding cross section has been removed in the new manuscript

Response: Thanks for your good advice. I have removed this sentence in the resubmitted manuscript. Thank you again.